# Student-, Study- and COVID-19-Related Predictors of Students’ Smoking, Binge Drinking and Cannabis Use before and during the Initial COVID-19 Lockdown in The Netherlands

**DOI:** 10.3390/ijerph19020812

**Published:** 2022-01-12

**Authors:** Kirsten J. M. van Hooijdonk, Milagros Rubio, Sterre S. H. Simons, Tirza H. J. van Noorden, Maartje Luijten, Sabine A. E. Geurts, Jacqueline M. Vink

**Affiliations:** Behavioural Science Institute, Radboud University, 6525 GD Nijmegen, The Netherlands; milagros.rubio@ru.nl (M.R.); sterre.simons@ru.nl (S.S.H.S.); tirza.vannoorden@ru.nl (T.H.J.v.N.); maartje.luijten2@ru.nl (M.L.); sabine.geurts@ru.nl (S.A.E.G.); jacqueline.vink@ru.nl (J.M.V.)

**Keywords:** COVID-19, pandemic, student, tobacco smoking, binge drinking, cannabis

## Abstract

Tobacco, alcohol and cannabis are commonly used among university students. However, student lives and their substance use have changed dramatically since the start of the COVID-19 pandemic. This study investigated the impact of COVID-19 on (trends in) weekly smoking, weekly binge drinking and weekly cannabis use in Dutch university students and investigated associated student-, study- and COVID-19-related characteristics. Between April and June 2020, several Dutch higher educational institutes invited their students to participate in an online survey. Data of 9967 students (M_age_ = 22.0 (SD = 2.6); N_female_ = 7008 (70.3%)) were available for analyses. Overall, weekly smoking remained stable (±11.5%), weekly binge drinking decreased (from 27.8% to 13.9%) and weekly cannabis use increased (from 6.7% to 8.6%). Male gender, not living with parents, being a bachelor student, having less financial resources and less adherence to the COVID-19 measures were found to increase the risk of substance use (before/during the first COVID-19 lockdown). Additionally, male gender, not living with parents, being a bachelor student, not being born in the Netherlands and having a student loan contributed to the likelihood of increased substance use during COVID-19. Patterns of characteristics contributing to the likelihood of decreased weekly substance use during COVID-19 were less clear. The risk factors male gender, not living with parents and being a bachelor student do not only contribute to the likelihood of using substances but also contribute to the likelihood of increased use during a lockdown. Prevention and intervention programs should especially target these risk groups.

## 1. Introduction

Before the COVID-19 pandemic, the prevalence of tobacco smoking, binge drinking and cannabis use among university students were relatively high compared to peers who are not attending higher education [1,2,3,4]. In the Netherlands, 23–29% of the students in higher education smoked occasionally, and 11–20% of the students smoked on a daily basis [5]. A large majority of Dutch university students consumed alcohol (92.8% lifetime use), and almost 30% were binge drinking more than once a month (in the past year) [6,7]. Furthermore, almost half of the Dutch university students (42%) had used cannabis at least once in their lives [7]. The Dutch prevalence of substance use among students is in line with international findings [8,9,10].

Since the start of the COVID-19 pandemic (March 2020), student lives have changed dramatically worldwide. Governments implemented measures to prevent the spread of the virus, e.g., social distancing, the closing of public spaces and the catering industry, and restrictions regarding indoor and team sports [11,12]. In many countries, schools and universities closed, and lockdowns were introduced. The closing of the university campus resulted in changed (online) teaching methods [13]. Consequently, many students faced challenges such as uncertainties about the future, loneliness, study delay, increased informal care responsibilities or illnesses of themselves, relatives or friends [14]. These challenges might have resulted in changes in smoking, binge drinking and cannabis use.

The impact of the first COVID-19 lockdown on students’ smoking, binge drinking and cannabis use was already studied in several international (university) student samples. Smoking did not change in German and Italian students [15,16]. In contrast, a decrease in tobacco smoking was reported in French students [17]. A decrease in overall alcohol consumption/binge drinking during the first COVID-19 lockdown was observed in many student samples [16,17,18,19,20,21,22]. The impact of the first COVID-19 lockdown on cannabis use was less clear, indicating either a decrease in French students [17] or no changes in German students [16].

Insights from before the COVID-19 pandemic showed that several student- and study-related factors play a role in students’ smoking, binge drinking and/or cannabis use. Being male, being a gender minority, being of white ethnicity, mental health problems, participation in extracurricular activities (e.g., student association), college residence (not living with parents) and being in their first year of college were associated with higher substance use [1,23,24]. Additionally, lower academic performance was associated with substance use, as well as higher perceived academic pressure and peer pressure [10,24,25,26,27].

Although these studies provide valuable insights, the amount of evidence regarding student- and study-related factors associated with substance use during the COVID-19 pandemic is limited. Up-to-date insights regarding the impact of the first COVID-19 lockdown on students’ substance use and associated characteristics are essential to provide adequate interventions and targeted services to (specific subgroups of) students and prepare for future lockdown situations.

According to our knowledge, no study exists that extensively investigated a wide variety of student- and study-related factors associated with (trends in) smoking, binge drinking and/or cannabis use in students in higher education, comparing the period before and during the first COVID-19 lockdown in the Netherlands (March 2020 to June 2020). In the current study, we aimed to (1) investigate changes in the prevalence of weekly smoking, weekly binge drinking and weekly cannabis use before and during the first COVID-19 lockdown; (2) explore which student-, study- and/or COVID-19-related characteristics are associated with weekly smoking, weekly binge drinking and weekly cannabis use both before and during the first COVID-19 lockdown; and (3) explore which pre-COVID-19 student-, and study-related characteristics contribute to changes in weekly smoking, weekly binge drinking or weekly cannabis use during the first COVID-19 lockdown in (a) non-users before COVID-19 and (b) weekly users before COVID-19.

## 2. Materials and Methods

The methods and data analyses were pre-registered in the Open Science Framework (OSF) [28].

### 2.1. Recruitment and Participants

This study is part of the COVID-19 International Student Well-Being Study (study protocol and questionnaire developed by Van de Velde et al. [29] University of Antwerp, Belgium). Several Dutch higher education institutes participated and invited their students via email to participate in an online survey (between 27 April 2020 and 7 July 2020). Data from students of Groningen University, Inholland University of Applied Sciences, Radboud University, Wageningen University and Research, Maastricht University and the University of Amsterdam were used for the current study (informed consent was obtained for all participants). Ethical approval was granted by the Ethical committee of the University of Antwerp and all Ethics Committees of participating Dutch institutes.

Before data cleaning, data of 14,675 students were available, including both university (research-oriented) and applied university (focused on practical skills) students, referred to as university students. Students who had not finished the survey (N = 3462), international students who moved back to their country of residence before or at the start of the COVID-19 pandemic (COVID-19 regulations differ per country; N = 766), participants aged ≥ 31 years old (older students might have a different lifestyle; N = 404) and participants who did not identify themselves as male or female were excluded from further analyses (the group of participants who reported their gender as “x” was too small to include; N = 76). In total, data of 9967 students were available for the main analyses (model A). The set of analyses described under model B was conducted in subgroups of non-users before COVID-19 for all three substances. The set of analyses described under model C was conducted in subgroups of weekly users before COVID-19 for all three substances.

### 2.2. Measures

Validated survey instruments were selected from previous international (student) surveys when available, and new survey items were developed when existing instruments were not suitable for questioning the COVID-19 pandemic. Details on survey topics are described in the study protocol [29]. Items included in this study are mentioned below.

For a substantial amount of the survey items, participants were asked to describe the situation before and during the COVID-19 regulations. “Before COVID-19” referred to the average situation during the month prior to the implementation of the first COVID-19 regulations. “During COVID-19” referred to the week prior to filling out the survey (i.e., during the first COVID-19 lockdown in The Netherlands).

#### 2.2.1. Dependent Variables

##### 2.2.1.1. Model A—Weekly Substance Use before/during the First COVID-19 Lockdown

In order to investigate weekly substance use before and during COVID-19, six outcome variables were created: weekly smoking before COVID-19 (1) and during COVID-19 (2); weekly binge drinking before COVID-19 (3) and during COVID-19 (4); weekly cannabis use before COVID-19 (5) and during COVID-19 (6). These variables were created based on questions regarding the frequency of tobacco smoking, binge drinking (drinking ≥6 glasses on a single occasion) and cannabis use before and during COVID-19. Response categories were dichotomised into “non-user” (0) (including (1) (almost) never, (2) less than once a week) versus “weekly user” (1) (including (3) once a week, (4) more than once a week and (5) (almost) daily). Participants who answered (6) “prefer not to say” either before or during COVID-19 were disregarded for the analysis of that particular substance.

##### 2.2.1.2. Model B—Non-Users before COVID-19

In order to investigate increases in weekly smoking, weekly binge drinking and weekly cannabis use in students who were non-users before COVID-19, dichotomous outcome variables were created for all three substances: “stable low” (0) versus “increase” (1), where “stable low” referred to being a non-user before and during COVID-19 and “increase” referred to moving from being a non-user before COVID-19 to weekly user during COVID-19.

##### 2.2.1.3. Model C—Weekly Users before COVID-19

In order to investigate decreases in weekly smoking, weekly binge drinking and weekly cannabis use in students who were weekly users before COVID-19, dichotomous outcome variables were created for all three substances: “stable high” (0) versus “decrease” (1), where “stable high” referred to being a weekly user before and during COVID-19 and “decrease” referred to moving from being a weekly user before COVID-19 to non-user during COVID-19.

#### 2.2.2. Independent Variables—Student-, Study- and COVID-19-Related Characteristics

Student-, study- and COVID-19-related characteristics were added in blocks: (1) demographics (gender, age, country of birth, educational level parents, relationship status during COVID-19 and living situation), (2) study-related information before and/or during COVID-19 (study program, the first year of education, the importance of study compared to other activities, paying tuition, financial resources, hours spent on study or job), (3) study attitude factors during COVID-19 (work attitude, participation in study activities and being in touch with fellow students), (4) study stress and well-being during COVID-19 (study-related concerns and depressive symptoms) and (5) other factors related to the COVID-19 situation (COVID-19 infection, adherence and worries). Detailed information on coding and scales is shown in Appendix A.

### 2.3. Statistical Analyses

Prevalence rates of weekly smoking, weekly binge drinking and weekly cannabis use before and during the first COVID-19 lockdown were analysed by using Chi-square tests (to test gender differences) and McNemar tests for paired observations (to explore differences within groups (men/women/total) over time).

Model A—Six multivariate logistic regression analyses were used to explore the associations between weekly smoking, weekly binge drinking or weekly cannabis use before or during the first COVID-19 lockdown on the one hand (see Section 2.2.1.1) and all independent variables together on the other hand (see Section 2.2.2). Independent variables were entered in the five blocks described above. Blocks 1–2 were added in the analyses before COVID-19, and all five blocks were added in the analyses during COVID-19. The backward selection was used within each step, only selecting significant variables (*p* < 0.05) for the next step in the model. In the final model, only variables with *p* < 0.0083 were deemed statistically significant (0.05/6; Bonferroni correction). The OR, *p*-value and explained variance (Nagelkerke R^2^) were reported. Additionally, to investigate the effect of combining multiple independent variables on the obtained associations, univariate logistic regression analyses (including gender and age as covariates) were performed to explore the individual relationships between the mean outcome variables and all separate independent variables (results are added to Appendix A).

Model B and C—Two multivariate logistic regression analyses were used to investigate whether pre-COVID-19 independent variables were associated with trends in weekly smoking, weekly binge drinking and weekly cannabis use during the first COVID-19 lockdown (see Section 2.2.1.2 and Section 2.2.1.3). In both models, blocks 1–2 were entered in steps. Measures during COVID-19 were not taken into account because we aimed to investigate which pre-COVID-19 factors would predict changes in substance use during COVID-19. The backward selection was used within each step, only selecting significant variables (*p* < 0.05) for the next step in the model. In the final models, only variables with *p* < 0.0167 were deemed statistically significant (0.05/3; Bonferroni correction). OR, *p*-values and NagelKerke R^2^ were reported.

## 3. Results

### 3.1. Participant Characteristics

The majority of the students were female, born in the Netherlands, had highly educated parents, enrolled in bachelor programs and were not first-year students (Table 1). A subsequent amount of the students moved back to their parents during the first COVID-19 lockdown, and an increase was observed in the number of students who indicated not having enough financial resources. As expected, hours spent on offline courses decreased while hours spent on online courses increased. Furthermore, at the time of completing the survey (April–July 2020), most participants had not been infected with COVID-19, and participants were more worried that someone in their personal network would become (re-)infected with COVID-19 or become severely ill from a COVID-19 (re-)infection than that this would happen to themselves.

### 3.2. Prevalence Rates Substance Use before and during COVID-19

No difference in the number of weekly smokers was observed when comparing before and during the first COVID-19 lockdown (Table 2). The number of weekly binge drinkers decreased (*p* < 0.001) while the number of weekly cannabis users increased (*p* < 0.001). Both before and during the first COVID-19 lockdown, there were more males than females who were weekly smokers, weekly binge drinkers or weekly cannabis users (*p* < 0.001).

### 3.3. Model A—Factors Associated with Substance Use before and/or during the First COVID-19 Lockdown

Overall, male gender (vs. female gender), not living with parents (vs. living with parents), being a bachelor student (vs. master student) and having less financial resources to cover monthly costs were associated with higher chances of weekly smoking, weekly binge drinking and weekly cannabis use (before/during COVID-19), see Table 3. Older age was associated with a higher chance of weekly smoking (before/during COVID-19) and a lower chance of weekly binge drinking (before COVID-19). Students not born in the Netherlands were more likely to be weekly smokers or weekly cannabis users (before/during COVID-19), while students born in the Netherlands were more likely to be weekly binge drinkers (only before COVID-19). Being in a complicated relationship (as compared to being in a steady relationship) increased the probability of weekly binge drinking and weekly cannabis use (during COVID-19). Students whose parents (partly) covered tuition of the current academic year were more likely to be weekly binge drinkers or weekly cannabis users (before COVID-19), and students who (partly) covered tuition via a bank or student loan were more likely to be weekly cannabis users (before/during COVID-19). Students who (partly) paid tuition themselves were less likely to be weekly smokers or weekly cannabis users (before/during COVID-19). Higher scores on depressive symptoms were associated with a higher chance of weekly smoking or weekly cannabis use (during COVID-19). Reporting stricter adherence to the implemented COVID-19 measures at the time of conducting the survey was associated with lower chances of weekly smoking, weekly binge drinking and weekly cannabis use (during COVID-19). Lastly, other measures such as the importance of study, hours spent on offline/online courses/study time/paid job, work attitude, concerns about study success, COVID-19 infection and worries were associated with some of the outcomes but not in a consistent pattern. Overall, the explained variance for the final models varied between 8.8 and 14.3% before COVID-19 and between 14.2% and 18.2% during COVID-19. The above-mentioned findings are in line with the results of the univariate logistic regression analyses (see Appendix A).

### 3.4. Factors Associated with Trends in Substance Use

For all three substances, the majority of the students were non-users both before and during the first COVID-19 lockdown (stable low) (Table 4). In the group of non-users before COVID-19, 2.7%, 6.2% and 3.4% increased to weekly smoking, weekly binge drinking and weekly cannabis use, respectively. In the group of weekly users before COVID-19, 20.0%, 66.1% and 19.5% became non-smokers, non-binge drinkers and non-cannabis users, respectively. The remaining students continued their weekly use during the lockdown.

#### 3.4.1. Model B—Non-Users before COVID-19

In the group of non-smokers, non-binge drinkers and non-cannabis users before COVID-19, male gender, not living with parents, bachelor programs, not being born in the Netherlands and having a student loan were associated with increased likelihood of becoming a weekly user during the first COVID-19 lockdown (especially weekly binge drinking and weekly cannabis use) (Table 5). Older age seems to increase the risk of becoming a weekly smoker and weekly binge drinker, whereas it seems to lower the risk of becoming a weekly cannabis user during COVID-19. Non-binge drinkers whose mother was highly educated were more likely to increase weekly binge drinking during COVID-19. Lastly, non-binge drinkers and non-cannabis users who report more time spent on a paid job before COVID-19 seem to be more likely to become weekly binge drinkers and weekly cannabis users during COVID-19. In contrast, non-cannabis users who report more personal study time before COVID-19 seem to be less likely to become weekly cannabis users during COVID-19. The explained variance for the final models varied from 4.6% for smoking to 6.1% for binge drinking, and 5.1% for cannabis use.

#### 3.4.2. Model C—Weekly Users before COVID-19

In these analyses, many of the variables were not significant and therefore excluded from the final models (Table 6). Sample sizes were small in de decrease groups for smoking (N = 228) and cannabis use (N = 130), which probably resulted in power issues. For weekly smokers and weekly binge drinkers before COVID-19, younger students were more likely to become non-smokers and non-binge drinkers during the first COVID-19 lockdown. Additionally, for weekly binge drinkers before COVID-19, female gender, having a highly educated father and living alone seems to increase the likelihood to become a non-binge drinker during COVID-19. Additionally, students with less financial resources and students who spent more time on a paid job were more likely to remain weekly binge drinkers during COVID-19. For weekly cannabis users, being in a bachelor program was associated with an increased likelihood of becoming a non-cannabis user during COVID-19. Explained variances for the final models were 2.1% for smoking, 10.3% for binge drinking and 3.2% for cannabis use.

## 4. Discussion

This study investigated the impact of the first COVID-19 lockdown on weekly substance use and associated student-, study- and COVID-19-related characteristics in Dutch university students.

The findings indicate clear risk factors for substance use among all three substances: male gender, not living with parents, being a bachelor student, having less financial resources and less adherence to the COVID-19 regulations. Other risk factors such as not being born in the Netherlands, not paying tuition themselves or parents paying intuition and depressive symptoms were associated with increased risk of using either one or two of the investigated substances. Additionally, clear overlap was observed in factors contributing to substance use before and during COVID-19. This indicates that characteristics contributing to substance use in university students are stable over time and are not so much influenced by unexpected stressful events such as the first lockdown. Additionally, a large overlap in identified risk factors (e.g., male gender, not living with parents, being a bachelor student) was observed in the cross-sectional analyses regarding substance use before and during COVID-19 (Model A) and in the analyses investigating changes in substance use during the first lockdown compared to before (Model B and C). This indicates that the identified characteristics are not only contributing to the likelihood of substance use but also contribute to changes in substance use. Striking findings both in line and in contrast with the previous literature are discussed below.

### 4.1. Comparison with Previous Studies

In line with other studies, we did not observe a change in smoking prevalence during the first COVID-19 lockdown [15,16]. This was expected as smoking is probably less driven by the social context and therefore less impacted by the COVID-19 measures. Moreover, in line with other studies, we observed a decrease in binge drinking/alcohol consumption prevalence during COVID-19 [16,17,18,19,20,21,22]. This decrease can be linked to the closing of all catering facilities, which limited the possibility for students to consume alcohol in social settings. Furthermore, we observed an increase in cannabis use prevalence during COVID-19, which is in contrast with other studies [16,17]. In the Netherlands, the recreational use and possession of cannabis are tolerated (more than in other countries), which might explain the increased use when students were restricted to their homes.

Insights regarding the role of gender in changes in substance use during the COVID-19 pandemic in students are inconsistent, possibly due to different methodological approaches, study samples or implemented COVID-19 measures. Observed changes in substance use suggested positive and negative effects for males on binge drinking and cannabis use [17], positive effects for females on drinking only [16] or no influence of gender on change in drinking behaviour [18]. Our findings suggested that being male is a risk factor for substance use before/during COVID-19 and also influences the likelihood to increase substance use (in non-users before COVID-19) or to remain a weekly binge drinker (in weekly binge drinkers before COVID-19).

In line with our findings, White et al. [18] showed that the number of drinking days, drinks per week and the maximum number of drinks per day were higher in students who lived with peers versus students who lived with parents before and/or during COVID-19. Tavolacci et al. [17] showed that not living with parents (both before and during COVID-19) was associated with negative changes in binge drinking during COVID-19. Our study supports the idea that not living with parents increases the likelihood of (increased) substance use. This was expected as living independently (and often in student accommodations) decreases parental supervision and increases social interaction, therefore facilitating the use of substances.

Previous studies conducted before COVID-19 showed that mental health or mood problems often co-occur with substance use (disorders) [30,31,32,33,34,35]. During the first COVID-19 lockdown, a German and a French study involving student populations showed that depressive symptoms were associated with unfavourable changes in health behaviour (including substance use) [16,17]. Our findings are in line with this observation as more depressive symptoms were associated with a higher risk of weekly smoking and weekly cannabis use. Other characteristics that were identified in this study to impact substance use before and/or during COVID-19, such as relationship status, having no financial resources and country of birth, could indirectly reflect mental health or mood problems. For relationship status, only the category “it’s complicated” was associated with higher risks of substance use. This is largely in line with the findings of Busse et al. [16]. Although speculative, uncertain relationship status or having no or not enough financial resources might affect mental health or mood and, therefore, substance use. For international students, it could be expected, especially during the COVID-19 pandemic, that less experienced social support from family and friends could also lead to mood problems or loneliness and explain the co-occurrence with increased binge drinking and cannabis use during COVID-19. In contrast, a German study did not find an association between increased substance use during the first COVID-19 lockdown and being an international student [16]. It should be noted that further research is needed to explore further the underlying mechanisms and the significance of the identified associations.

### 4.2. Strengths and Limitations

To the best of our knowledge, this is the first study to examine the association between (trends in) substance use and student-, study- and COVID-19-related characteristics in Dutch university students, investigating a comprehensive set of factors both cross-sectionally and over time (retrospective). This broader perspective is important to understand the mechanisms of students’ substance use (and the relative importance of contributing factors).

Several limitations need to be considered. The study sample consists of different subsamples of Dutch (applied) university students. Although this makes it possible to generalise the results to different Dutch student populations, it limits the generalisability to international student populations due to differences in the education system and COVID-19 regulations. Selection bias might be playing a role if students with extremely high levels of substance use or poor well-being did not participate in the study. However, a major impact is not expected, as prevalence rates of smoking, binge drinking and cannabis use align with earlier research in Dutch university students [5,36]. Next, students needed to recall information from the month prior to the implementation of the first COVID-19 regulations. Although this might have introduced inaccuracies, it was expected that students could accurately recall the period before the first COVID-19 lockdown due to the large contrast in context. Next, variables in our regression models were selected based on *p* < 0.05. It appeared that after adding variables from additional blocks, some variables from earlier added blocks did not remain significant in the final model. Therefore, the order of adding the variables influenced the obtained results.

### 4.3. Future Research and General Implications

(1) Follow-up research should pay attention to high substance use risk groups (e.g., males, students not living with parents, bachelor students, students with less financial resources, students who adhere less to the COVID-19 regulations, students not born in the Netherlands, students not paying tuition themselves or whose parents paid tuition and students with depressive symptoms) and how substance use within these groups will further develop during the current continuation of the pandemic. (2) Despite the general not so alarming trends in prevalence rates, there are subgroups of students who (re)started using substances or who further increased their weekly use during the COVID-19 pandemic. Extra support and further research are needed for these groups. (3) In this study, some risk factors were identified who contribute to the use of all three investigated substances, while also risk factors were identified who only contribute to the use of one or two substances only. Future research should focus on the unique mechanisms and involved characteristics per substance. (4) Polysubstance use should be taken into account since it is known that the use of one substance may increase the likelihood of using another [37]. Additionally, although tobacco, alcohol and cannabis are the most prevalently used substances among students, future studies should also include the use of other (popular) substances such as ecstasy/MDMA [7]. (5) As the COVID-19 pandemic is continuing, monitoring substance use among university students during different phases of the pandemic (partial lockdowns, changing regulations) and after the pandemic will remain important, as a continuation of lockdown measures could increase substance use in the identified risk groups while loosening of measures might go together with outbursts of hazardous substance use and “compensatory behaviour” among (other subgroups of) students.

### 4.4. Recommendations for Policymakers and Practice

This study provides valuable insights for public authorities and higher education institutes, especially since the COVID-19 pandemic is still ongoing, and students, higher education institutes and policymakers will also have to cope with its consequences in the near future. Public authorities might want to pay extra attention to high-risk groups of substance use when developing/improving interventions and policies addressing substance use among students in higher education. Additionally, public authorities should ensure that higher education institutes have adequate resources to train their staff and provide effective and accessible services to help students adopt healthy lifestyles. Higher education institutes should ensure that support services (e.g., counselling, guidance and interventions to reduce hazardous substance use) are available and that these services are accessible and user-friendly. Communication towards students about available services and practicalities should be provided in a clear and accessible way. Lastly, higher education institutes should be aware that there are different subgroups of students who are at elevated risk of using substances and that these students might need to be actively offered extra support. This is essential, not only during the COVID-19 pandemic but also when regulations are loosened, as supporting these students will contribute to the prevention or decrease in hazardous substance use (and related consequences) in the future.

## 5. Conclusions

This study makes a substantial contribution to the knowledge on the underlying mechanisms of substance use and the identification of substance use high-risk groups both before and during the first COVID-19 lockdown. These valuable new insights on how students cope during a global health crisis can directly be applied by public authorities and higher education institutes by enhancing implemented policies, interventions and support services. This is of utmost relevance as the COVID-19 pandemic is still ongoing, and new lockdowns/restrictions are still in place.

## Figures and Tables

**Table 1 ijerph-19-00812-t001:** Participant characteristics.

	Before	During
	N (%) or Mean (SD; N)	N (%) or Mean (SD; N)
1-Demographics		
Gender		
Female	7008 (70.3%)	-
Male	2959 (29.7%)	-
Age (years)	22.0 (2.6; 9967)	-
Country of birth		
Netherlands	8336 (83.6%)	-
Other	1631 (16.4%)	-
Education mother		
(Less than) secondary	4302 (44.4%)	-
Higher education	5396 (55.6%)	-
Education father		
(Less than) secondary	3860 (40.3%)	-
Higher education	5725 (59.7%)	-
Relationship status		
Steady relationship	-	4527 (45.4%)
Single	-	5035 (50.5%)
Complicated	-	405 (4.1%)
Living situation		
With parent(s)	3020 (30.3%)	5100 (51.2%)
Student hall	1135 (11.4%)	656 (6.6%)
With others	3716 (37.3%)	2470 (24.8%)
Alone	1679 (16.8%)	1124 (11.3%)
Other	417 (4.2%)	617 (6.2%)
2-Study-related information before and/or during COVID-19		
Study program		
Bachelor	7178 (72.3%)	-
Master	2744 (27.7%)	-
First year		
Yes	1786 (18.0%)	-
No	8316 (82.0%)	-
Importance study—other activities		
More	4104 (41.2%)	-
Equally	5350 (53.7%)	-
Less	513 (5.1%)	-
Tuition current year paid by…		
Parents (no/yes)	4452 (44.7%)/5515 (55.3%)	-
Myself (no/yes)	7448 (74.7%)/2519 (25.3%)	-
Loan (no/yes)	7013 (70.4%)/2954 (29.6%)	-
Scholarship (no/yes)	9560 (95.9%)/407 (4.1%)	-
No sufficient financial resources (range: 1–5) ^a^	1.8 (0.7; 9967)	2.2 (1.1; 9967)
How many hours spent on…		
Offline courses	14.0 (11.0; 7284)	1.9 (6.6; 7298)
Online courses	2.2 (4.4; 7284)	8.8 (9.3; 7298)
Study time	13.3 (10.6; 7284)	15.1 (12.9; 7298)
Paid job	7.9 (9.2; 7284)	5.3 (9.4; 7298)
3-Study attitude factors during COVID-19		
I work hard to succeed in my studies and spend a sufficient amount of time (range: 1–5)	-	3.6 (1.1; 9967)
Usually, I participate in all study activities (range: 1–5)	-	3.8 (1.1; 9967)
Being in touch with my fellow students helps me to perform well (range: 1–5)	-	3.5 (1.0; 9967)
4-Study stress and well-being during COVID-19		
Workload increased (range: 1–5)	-	3.3 (1.2; 9967)
Concerned study success (range: 1–5)	-	3.3 (1.3; 9967)
Change in teaching causing stress (range: 1–5)	-	3.5 (1.2; 9967)
Depressive symptoms (range: 0–24)	-	10.2 (4.8; 9967)
5-Other factors related to the COVID-19 situation		
COVID-19 infection		
No	-	8580 (86.1%)
Yes, confirmed by test	-	18 (0.2%)
Yes, health care provider	-	147 (1.5%)
I think so, but not confirmed	-	1217 (12.2%)
Adhere to COVID-19 measures (range: 0–10)	-	7.8 (1.6; 9967)
COVID-19 related worries (range: 0–10)		
(Re-)infection self	-	3.8 (2.5; 9962)
Severely ill (re-)infection self	-	3.5 (2.6; 9962)
(Re-)infection personal network	-	6.8 (2.2; 9967)
Severely ill (re-)infection personal network	-	7.1 (2.3; 9967)
Sufficient medical supplies	-	5.0 (2.6; 9967)

Note: Before refers to the month prior to the implementation of the first COVID-19 regulations. During refers to a week prior to filling out the survey, during the first COVID-19 lockdown (April–July 2020). ^a^ Higher scores indicate fewer financial resources.

**Table 2 ijerph-19-00812-t002:** Prevalence of weekly smoking, weekly binge drinking and weekly cannabis use both before and during the first COVID-19 lockdown.

	Before			During			Difference Before vs. During ^$^		
	Women	Men	Total	Women	Men	Total	Women	Men	Total
Weekly smoking									
%	9.8% *	15.6% *	11.5%	9.8% *	15.8% *	11.6%	*p* = 1.000	*p* = 0.645	*p* = 0.816
N user/total	685/6983	460/2944	1145/9927	684/6974	462/2931	1146/9905	N = 6973	N = 2928	N = 9901
Weekly binge drinking									
%	23.8% *	37.3% *	27.8%	10.9% *	21.0% *	13.9%	***p* < 0.001**	***p* < 0.001**	***p* < 0.001**
N user/total	1666/7003	1101/2955	2767/9958	761/7004	619/2954	1380/9958	N = 7003	N = 2952	N = 9955
Weekly cannabis use									
%	4.1% *	12.9% *	6.7%	6.1% *	14.6% *	8.6%	***p* < 0.001**	***p* < 0.001**	***p* < 0.001**
N user/total	290/7000	379/2943	669/9943	424/6999	430/2944	854/9943	N = 6999	N = 2941	N = 9940

Note: Before refers to the month prior to the implementation of the first COVID-19 regulations. During refers to a week prior to filling out the survey, during the first COVID-19 lockdown (April–July 2020). * Significant sex differences in prevalence within group (Chi-square test, *p* < 0.001 for dichotomous variables). ^$^ McNemar test for paired observations, only including participants with data both before and during the first COVID-19 lockdown. N per analysis is provided in the table and significant differences (*p* < 0.001) are presented in bold.

**Table 3 ijerph-19-00812-t003:** Student-, study- and COVID-19-related factors associated with weekly smoking, weekly binge drinking or weekly cannabis use before and during the first COVID-19 lockdown (Model A) ^#^.

		Weekly Smoking				Weekly Binge Drinking				Weekly Cannabis Use			
		Before (N = 6859)		During (N = 6850)		Before (N = 6875)		During (N = 6884)		Before (N = 6864)		During (N = 6872)	
1-Demographics		OR	*p*	OR	*p*	OR	*p*	OR	*p*	OR	*p*	OR	*p*
Gender	Male */Female	**0.58**	<0.001	**0.59**	<0.001	**0.46**	<0.001	**0.46**	<0.001	**0.27**	<0.001	**0.39**	<0.001
Age		**1.09**	<0.001	**1.10**	<0.001	**0.87**	<0.001	-	-	-	-	0.96	0.094
Country of birth	Netherlands */Other	**2.06**	<0.001	**1.63**	<0.001	**0.44**	<0.001	0.76	0.022	**2.72**	<0.001	**1.88**	<0.001
Education mother	(Less than) secondary */higher education	-	-	0.96	0.660	-	-	-	-	1.23	0.068	-	-
Education father	(Less than) secondary */higher education	-	-	-	-	-	-	0.86	0.055	-	-	-	-
Relationship status	Steady relationship *			**1**	0.008			**1**	0.007			**1**	0.003
	Single			0.90	0.260			1.08	0.361			0.91	0.383
	Complicated			1.56	0.013			**1.69**	0.002			**1.75**	0.004
Living situation	With parent(s) *	**1**	0.001	**1**	<0.001	**1**	<0.001	**1**	<0.001	**1**	<0.001	**1**	<0.001
	Student hall	1.09	0.540	**1.70**	0.001	**1.79**	<0.001	**2.76**	<0.001	**1.99**	<0.001	**3.31**	<0.001
	With others	**1.49**	<0.001	**2.20**	<0.001	**2.40**	<0.001	**2.82**	<0.001	**2.59**	<0.001	**3.64**	<0.001
	Alone	1.34	0.022	**2.36**	<0.001	**1.55**	<0.001	**1.87**	<0.001	**2.13**	<0.001	**3.76**	<0.001
	Other	1.46	0.052	**2.08**	<0.001	1.26	0.171	**2.46**	<0.001	**2.30**	0.004	**3.01**	<0.001
Nagelkerke R square		3.3%		6.0%		9.6%		7.9%		9.9%		11.6%	
2-Study-related information before and/or during COVID-19		OR	*p*	OR	*p*	OR	*p*	OR	*p*	OR	*p*	OR	*p*
Study program	Bachelor */Master	**0.49**	<0.001	**0.50**	<0.001	-	-	**0.75**	0.001	**0.68**	0.002	0.82	0.117
First year	Yes */no	-	-	0.80	0.064	-	-	-	-	-	-	0.70	0.010
Importance study—other activities	More *	-	-	-	-	**1**	<0.001	**1**	<0.001	**1**	<0.001	1	0.074
	Equally	-	-	-	-	**1.72**	<0.001	**1.39**	<0.001	1.28	0.036	1.20	0.091
	Less	-	-	-	-	**2.05**	<0.001	1.44	0.019	**2.34**	<0.001	1.49	0.038
Tuition current year paid by…	Parents no */yes	-	-	-	-	**1.21**	0.004	-	-	**1.58**	0.004	1.37	0.027
	Myself no */yes	**0.66**	<0.001	**0.69**	<0.001	0.84	0.019	0.82	0.020	**0.63**	0.002	**0.68**	0.004
	Loan no */yes	1.19	0.045	-	-	-	-	-	-	**1.65**	0.001	**1.49**	0.004
	Scholarship no */yes	0.56	0.017	0.60	0.042	0.65	0.018	0.60	0.030	-	-	-	-
No financial resources ^a^		**1.20**	<0.001	**1.21**	<0.001	**1.19**	<0.001	**1.30**	<0.001	-	-	**1.24**	<0.001
How many hours spent on…	Offline course	0.99	0.034	-	-	**0.99**	<0.001	-	-	-	-	0.99	0.102
	Online course	-	-	**0.98**	0.002	**1.02**	0.002	**0.99**	0.002	-	-	-	-
	Study time	**0.99**	<0.001	1.00	0.164	**0.99**	<0.001	0.99	0.021	-	-	-	-
	Paid job	**1.03**	<0.001	1.01	0.011	**1.02**	<0.001	1.01	0.046	1.01	0.012	-	-
Nagelkerke R square		+ 5.5%		+4.8%		+4.7%		+4.8%		+2.7%		+3.2%	
3-Study attitude factors during COVID-19				OR	*p*			OR	*p*			OR	*p*
I work hard to succeed in my studies and spend a sufficient amount of time				0.92	0.044			-	-			**0.89**	0.005
Usually, I participate in all study activities				-	-			0.93	0.036			-	-
Being in touch with my fellow students helps me to perform well				-	-			1.08	0.058			-	-
Nagelkerke R square				+0.3%				+0.3%				+0.5%	
4-Study stress and well-being during COVID-19				OR	*p*			OR	*p*			OR	*p*
Workload increased				1.09	0.024			-	-			-	-
Concerned study success				-	-			**1.12**	<0.001			-	-
Change in teaching causing stress				-	-			-	-			-	-
Depressive symptoms				**1.04**	<0.001			-	-			**1.03**	0.004
Nagelkerke R square				+0.8%				+0.2%				+0.2%	
5-Other factors related to the COVID-19 situation				OR	*p*			OR	*p*			OR	*p*
COVID-19 infection	No *			-	-			-	-			**1**	<0.001
	Yes, confirmed by test			-	-			-	-			<0.001	0.998
	Yes, health care provider			-	-			-	-			1.82	0.055
	I think so, but not confirmed			-	-			-	-			**1.61**	<0.001
Adhere to COVID-19 measures				**0.82**	<0.001			**0.75**	<0.001			**0.84**	<0.001
COVID-19 related worries	(Re-)infection self			-	-			0.97	0.047			-	-
	Severely ill (re-)infection self			-	-			-	-			**0.94**	0.003
	(Re-)infection personal network			-	-			-	-			-	-
	Severely ill (re-)infection personal network			**1.07**	<0.001			-	-			1.05	0.028
	Sufficient medical supplies			**0.96**	0.008			-	-			-	-
Nagelkerke R square				+2.3%				+5.0%				+2.5%	
Total Nagelkerke R square		8.8%		14.2%		14.3%		18.2%		12.6%		18.0%	

Note: Multivariate associations between the dichotomous main outcome variables (weekly smoking before/during, weekly binge drinking before/during, weekly cannabis use before/during) and demographics (block 1), study-related information (block 2), study attitude factors (block 3), study stress and well-being due to COVID-19 (block 4) and other factors related to the COVID-19 situation (block 5). All independent variables were entered per block, with two blocks (block 1, 2) in the analyses with substance use before COVID-19, and three additional blocks (block 3, 4, 5) were added in the analyses with substance use during COVID-19. “Before” refers to the month prior to the implementation of the first COVID-19 regulations. “During” refers to a week prior to filling out the survey, during the first COVID-19 lockdown (April–July 2020). * Reference category predictor. ^#^ Significant predictors (*p* < 0.0083) are presented in bold (Bonferroni correction 0.05/6). ^a^ Higher scores indicate fewer financial resources.

**Table 4 ijerph-19-00812-t004:** Overview of trends in substance use during the first COVID-19 lockdown in non-users and weekly users before COVID-19.

		Smoking		Binge Drinking		Cannabis Use	
Before	Trend	N	%	N	%	N	%
Non-users	Stable low	8528	97.3	6745	93.8	8956	96.6
	Increase to weekly use	234	2.7	443	6.2	316	3.4
**Total**		8762	100.0	7188	100.0	9272	100.0
Weekly users	Stable high	911	80.0	937	33.9	538	80.5
	Decrease to non-use	228	20.0	1830	66.1	130	19.5
**Total**		1139	100.0	2767	100.0	668	100.0

Note: Non-users before COVID-19 were divided into (1) stable low, which refers to substance use (smoking, binge drinking or cannabis use) less than weekly, both before and during COVID-19, or (2) increase, which refers to moving from less than weekly use to weekly use or more. Weekly users before COVID-19 were divided into (1) stable high, which refers to weekly substance use or more, both before and during COVID-19, or (2) decrease, which refers referring to moving from weekly use or more to less than weekly use.

**Table 5 ijerph-19-00812-t005:** Trends in substance use during the first COVID-19 lockdown in non-users before COVID-19: stable low vs. increase (Model B) ^#^.

		Smoking Stable Low vs. Increase (N = 6097)		Binge Drinking Stable Low vs. Increase (N = 4885)		Cannabis Use Stable Low vs. Increase (N = 6453)	
1-Demographics		OR	*p*	OR	*p*	OR	*p*
Gender	Male */Female	**0.57**	0.001	**0.73**	0.022	**0.58**	<0.001
Age		**1.14**	0.001	**1.08**	0.003	**0.92**	0.010
Country of birth	Netherlands */Other	-	-	**1.54**	0.006	**1.68**	0.010
Education mother	(Less than) secondary */higher education	-	-	**1.44**	0.006	-	-
Education father	(Less than) secondary */higher education	-	-	-	-	0.80	0.130
Living situation	With parent *	**1**	<0.001	**1**	0.002	**1**	<0.001
	Student hall	**2.38**	0.003	**1.78**	0.009	**2.91**	<0.001
	With others	**2.75**	<0.001	**2.01**	<0.001	**2.87**	<0.001
	Alone	**1.99**	0.017	**1.71**	0.009	**2.51**	<0.001
	Other	0.98	0.965	1.75	0.052	1.46	0.404
Nagelkerke R square		3.0%		2.9%		3.0%	
2- Study-related information before COVID-19		OR	*p*	OR	*p*	OR	*p*
Study program	Bachelor */Master	**0.51**	0.002	**0.64**	0.004	-	-
First year	Yes */no	-	-	-	-	-	-
Importance study—other activities	More *	-	-	1	0.021	-	-
	Equally	-	-	**1.38**	0.015	-	-
	Less	-	-	1.71	0.043	-	-
Tuition current year paid by…	Parents no */yes	1.43	0.050	-	-	-	-
	Myself no */yes	-	-	-	-	-	-
	Loan no */yes	-	-	**1.37**	0.016	**1.52**	0.006
	Scholarship no */yes	-	-	-	-	-	-
No financial resources ^a^		-	-	-	-	-	-
How many hours spent on…	Offline course	1.02	0.027	-	-	-	-
	Online course	-	-	-	-	-	-
	Study time	-	-	0.99	0.051	**0.98**	0.013
	Paid job	1.02	0.021	**1.03**	<0.001	**1.03**	<0.001
Nagelkerke R square		+1.6%		+3.2%		+2.1%	
Total Nagelkerke R square		4.6%		6.1%		5.1%	

Note: Multivariate associations between the dichotomous trend variables (smoking, binge drinking and cannabis use stable low vs. increase) and demographics (block 1) and study-related information before COVID-19 (block 2). All independent variables were entered per block. * Reference category predictor. ^#^ Significant predictors (*p* < 0.0167) are presented in bold (Bonferroni correction 0.05/3). ^a^ Higher scores indicate fewer financial resources.

**Table 6 ijerph-19-00812-t006:** Trends in substance use during the first COVID-19 lockdown in weekly users before COVID-19: stable high vs. decrease (Model C) ^#^.

		Smoking Stable High vs. Decrease (N = 742)		Binge Drinking Stable High vs. Decrease (N = 1988)		Cannabis Use Stable High vs. Decrease (N = 409)	
1-Demographics		OR	*p*	OR	*p*	OR	*p*
Gender	Male */Female	-	-	**2.05**	<0.001	-	-
Age		**0.89**	0.002	**0.86**	<0.001	-	-
Country of birth	Netherlands */Other	-	-	-	-	-	-
Education mother	(Less than) secondary */higher education	-	-	-	-	-	-
Education father	(Less than) secondary */higher education	-	-	**1.31**	0.009	-	-
Living situation	With parent *	-	-	1	0.023	-	-
	Student hall	-	-	0.91	0.565	-	-
	With others	-	-	0.96	0.752	-	-
	Alone	-	-	**1.52**	0.015	-	-
	Other	-	-	0.68	0.202	-	-
Nagelkerke R square		2.1%		7.6%		0.0%	
2-Study-related information before COVID-19		OR	*p*	OR	*p*	OR	*p*
Study program	Bachelor */Master	-	-	1.38	0.021	**0.38**	0.010
First year	Yes */no	-	-	-	-	-	-
Importance study—other activities	More *	-	-	-	-	-	-
	Equally	-	-	-	-	-	-
	Less	-	-	-	-	-	-
Tuition current year paid by…	Parents no */yes	-	-	-	-	-	-
	Myself no */yes	-	-	-	-	-	-
	Loan no */yes	-	-	-	-	-	-
	Scholarship no */yes	-	-	-	-	-	-
No financial resources ^a^		-	-	**0.79**	<0.001	-	-
How many hours spent on…	Offline course	-	-	-	-	-	-
	Online course	-	-	-	-	-	-
	Study time	-	-	-	-	-	-
	Paid job	-	-	**0.97**	<0.001	-	-
Nagelkerke R square		+0.0%		+2.7%		+3.2%	
Total Nagelkerke R square		2.1%		10.3%		3.2%	

Note: Multivariate associations between the dichotomous trend variables (smoking, binge drinking and cannabis use stable high vs. decrease) and demographics (block 1) and study-related information before COVID-19 (block 2). All independent variables were entered per block. * Reference category predictor. ^#^ Significant predictors (*p* < 0.0167) are presented in bold (Bonferroni correction 0.05/3). ^a^ Higher scores indicate fewer financial resources.

## Data Availability

The data presented in this study are available on request. The data are not publicly available because they are part of a longitudinal (ongoing) project of the Radboud University.

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
