# Peer review of "Student-, Study- and COVID-19-Related Predictors of Students’ Smoking, Binge Drinking and Cannabis Use before and during the Initial COVID-19 Lockdown in The Netherlands"

_ijerph, 2022, doi:10.3390/ijerph19020812_

Round 1

Reviewer 1 Report

Review

"Student-, study- and COVID-19-related predictors of students’ smoking, binge drinking and cannabis use before and during the initial COVID-19 lockdown in the Netherlands"

This study examines the impact of COVID-19 on (trends in) weekly smoking, weekly binge drinking and weekly cannabis use in Dutch university students and investigated associated student-, study- and COVID-19-related characteristics, an important and topical issue.

The topic has become even more important due to the re-eruption of COVID-19 and the assessment that policymakers and inhabitants will have to cope with it for much time to come. Also, the study and its unique findings may break new ground relative to the existing research literature on coping at times of health-related crisis and may offer a unique and practical angle of observation for the matter investigated. However, several remarks are in order:

  1. Add to the paragraph three to four of the most recent journal references in the introduction starting from lines 39 to 47.
  2. The chapter dealing with recommendations for future research and implications should be broader. Future research is clear, but the context for policy recommendations is insufficiently developed. Further recommendations should be considered that will support the findings of the study, in relation to the Covid epidemic. These issues are increasingly important mainly due to the continuation of the epidemic and the threats of further closures. The recommendations, as they now appear in the article, are too comprehensive and too few and do not sufficiently cover the findings in the existing context.
  3. The same is true with the conclusions chapter. It is very limited and does not relate to the significance of the current health situation in the context of the materials and their usage. You should consider inserting the conclusions chapter after paragraph d and before e.
    Future research and implications. A deeper development of the conclusions may substantiate the policy recommendations.

In sum, the topic of this article is very important and is meaningful for the expansion and enrichment of research literature in this field. Once minor changes are made in accordance with the foregoing, the article may be considered for publication in this journal.

Author Response

January, 6th 2022

Dear editors and reviewers,

Thank you for considering our manuscript ‘Student-, study- and COVID-19-related predictors of students’ smoking, binge drinking and cannabis use before and during the initial COVID-19 lockdown in the Netherlands’ for publication in the International Journal of Environmental Research and Public Health.

We would like to thank the reviewers for their kind words and helpful comments on our paper. We have revised the article and hope it is now suitable for publication in the International Journal of Environmental Research and Public Health. Please find below a detailed response to the feedback of the reviewers (changes in the manuscript are highlighted in track changes).

Best wishes, also on behalf of the co-authors,

Kirsten van Hooijdonk, MSc

Response to Reviewer 1 Comments

Point 1: Add to the paragraph three to four of the most recent journal references in the introduction starting from lines 39 to 47.

Response 1: To support the paragraph on implemented COVID measures, consequences for student lives and the faced challenges, the following references have been added:

  • Implemented COVID-19 measures:
    • (11) Cowling, B.J.; Aiello, A.E. Public Health Measures to Slow Community Spread of Coronavirus Disease 2019. The Journal of Infectious Diseases 2020, 221, 1749-1751, doi:10.1093/infdis/jiaa123. (line 42)
    • (12) ECDC. Guidelines for non-pharmaceutical interventions to reduce the impact of COVID-19 in the EU/EEA and the UK. 24 September 2020. ; 2020. (line 42)
  • Impact on student lives:
    • (13) Doolan, K.; Barada, V.; Buric, I.; Krolo, K.; Tonkovic, Ž. Student Life during the COVID-19 Pandemic Lockdown: Europe-Wide Insights; Brussels: European Students’ Union: 2021. (line 44)
  • Faced challenges:
    • (14) Aristovnik, A.; Keržič, D.; Ravšelj, D.; Tomaževič, N.; Umek, L. Impacts of the COVID-19 Pandemic on Life of Higher Education Students: A Global Perspective. Sustainability 2020, 12, 8438, doi:https://doi.org/10.3390/su12208438. (line 46)

Point 2:  The chapter dealing with recommendations for future research and implications should be broader. Future research is clear, but the context for policy recommendations is insufficiently developed. Further recommendations should be considered that will support the findings of the study, in relation to the Covid epidemic. These issues are increasingly important mainly due to the continuation of the epidemic and the threats of further closures. The recommendations, as they now appear in the article, are too comprehensive and too few and do not sufficiently cover the findings in the existing context.

Response 2: To broaden the chapter on future research and implications, we have split the paragraph “4.3 Future research and implications” into two separate paragraphs “4.3 Future research and general implications” and “4.4 Recommendations for policymakers and practice”. Paragraph “4.3 Future research and general implications” has been restructured and linked to the ongoing COVID-19 pandemic (lines 411-442). A new paragraph “4.4 Recommendations for policymakers and practice” has been written in which the findings are connected to practice and the ongoing COVID-19 pandemic (lines 444-461).

Point 3: The same is true with the conclusions chapter. It is very limited and does not relate to the significance of the current health situation in the context of the materials and their usage. You should consider inserting the conclusions chapter after paragraph d and before e.

Future research and implications. A deeper development of the conclusions may substantiate the policy recommendations.

Response 3: We have started the discussion section with a summary of the findings and with the most important conclusions. In the remaining part of the discussion, we discuss these findings, mention strengths and limitations and describe future research, implications and recommendations for policymakers and practice. The final paragraph is meant as a short concluding summary. To make this more clear, we have now changed the header name and made the link to the ongoing COVID-19 pandemic (lines 462-472).

Reviewer 2 Report

Thank you for the opportunity to contribute to the peer review process for the original study submission manuscript entitled “Student-, study- and COVID-19-related predictors of students’ 2 smoking, binge drinking and cannabis use before and during 3 the initial COVID-19 lockdown in the Netherlands”. The manuscript is interesting, well written and points out relevant issues.

The most important issue is related to Table 2:

Please, check the numbers of Women/Men “Before” and “During”. For McNemar test you considered paired observations, although totals of women and men are not equal. As the differences are too small, I suggest keeping just those with no missing information – weekly smoking, or changed?? (Binge drinking – Women/Men Before=7003/2955 and During=7004/2954 Cannabis use – Women/Men Before=7000/2943 and During 6999/2944).

Minor issues:

Line 119 – amend the number of the item (2.2.1.1, not 2.1.1.1)

Line 131 – amend the number of the item (2.2.1.2, not 2.1.1.2)

Author Response

January, 6th 2022

Dear editors and reviewers,

Thank you for considering our manuscript ‘Student-, study- and COVID-19-related predictors of students’ smoking, binge drinking and cannabis use before and during the initial COVID-19 lockdown in the Netherlands’ for publication in the International Journal of Environmental Research and Public Health.

We would like to thank the reviewers for their kind words and helpful comments on our paper. We have revised the article and hope it is now suitable for publication in the International Journal of Environmental Research and Public Health. Please find below a detailed response to the feedback of the reviewers (changes in the manuscript are highlighted in track changes).

Best wishes, also on behalf of the co-authors,

Kirsten van Hooijdonk, MSc

Response to Reviewer 2 Comments

Point 1: Please, check the numbers of Women/Men “Before” and “During”. For McNemar test you considered paired observations, although totals of women and men are not equal. As the differences are too small, I suggest keeping just those with no missing information – weekly smoking, or changed?? (Binge drinking – Women/Men Before=7003/2955 and During=7004/2954 Cannabis use – Women/Men Before=7000/2943 and During 6999/2944).

Response 1: For most participants data is available on weekly smoking, weekly binge drinking and weekly cannabis both before and during the first COVID-19 lockdown. However, a very small group of the participants lack data either before or during the first COVID-19 lockdown which results in the total number of women and men before and during the first COVID-19 pandemic not being equal. We understand this might be causing confusion for the reader. Therefore, we suggest to include N per analysis in the table (in the column “Difference before vs. during”) and adding an extra line of explanation regarding the McNemar test in the note below the table (lines 209-213).

Point 2: Line 119 – amend the number of the item (2.2.1.1, not 2.1.1.1)

Response 2: This inaccuracy has been corrected (track changes line 120).

Point 3: Line 131 – amend the number of the item (2.2.1.2, not 2.1.1.2)

Response 3: This inaccuracy has been corrected (track changes line 132).

Reviewer 3 Report

The authors explored part of the COVID-19 International Student Well-Being Study, a so far neglected area of possible pandemics/lockdown related problems, that of substance use/abuse in students as a special at-risk group. They identified risks factors (such as male gender, not living with parents, being a bachelor students, having less financial resources and less adherence to the COVID-19 regulations), factors not only contributing to the likelihood of substance use, but also contribute to changes in substance use.

The subject is relevant, methodology and the general style and format adequate. While style is in general good, a final check by a native speaker might be helpful.

Conclusions for improvement of the identified risks might be discussed in more detail. Information on the (multi-ethnic) background and its relevance is very short and might be extended, though the results on "born/not born in the Netherlands) is a good start.

Minor issues:

Tables 1-2 might be referred to and mentioned in more detail in the text.

A few unclear lines, Line 32 applied (?) (shortly explain the Dutch educational system here)

254: the majority of the students remained their use stable low (in ..?)

Author Response

January, 6th 2022

Dear editors and reviewers,

Thank you for considering our manuscript ‘Student-, study- and COVID-19-related predictors of students’ smoking, binge drinking and cannabis use before and during the initial COVID-19 lockdown in the Netherlands’ for publication in the International Journal of Environmental Research and Public Health.

We would like to thank the reviewers for their kind words and helpful comments on our paper. We have revised the article and hope it is now suitable for publication in the International Journal of Environmental Research and Public Health. Please find below a detailed response to the feedback of the reviewers (changes in the manuscript are highlighted in track changes).

Best wishes, also on behalf of the co-authors,

Kirsten van Hooijdonk, MSc

Response to Reviewer 3 Comments

Point 1: Conclusions for improvement of the identified risks might be discussed in more detail. Information on the (multi-ethnic) background and its relevance is very short and might be extended, though the results on "born/not born in the Netherlands) is a good start.

Response 1: Due to the amount of identified risk factors, we decided to keep the recommendations part concise and general, therefore not providing recommendations per identified risk factor. However, we understand information on some of the identified risk factors (e.g. background) was limited. Therefore, we extended the related information in lines 362-386.

Point 2: Tables 1-2 might be referred to and mentioned in more detail in the text.

Response 2: Table 1 is referred to in line 189 and Table 2 is referred to in line 204. Due to the word limit and priority to give more attention to the results of the multivariate logistic regression analyses (models A-C), we aimed to keep the paragraphs explaining the results of Table 1 and Table 2 concise by only pointing out the most striking results.

Point 3: A few unclear lines, Line 32 applied (?) (shortly explain the Dutch educational system here)

Response 3: We understand that the term “applied” might introduce some confusion among the readers. In the Netherlands, higher education consists of university and applied university students. As this term might be more widely used, we have changed “(applied) university students” into “students in higher education” (lines 32/33 and line 72). Furthermore, (to not disturb the flow in the introduction), we have added a few words of explanation to the method section (section 2.1), where it is mentioned that the study sample consists of both university and applied university students (lines 97/98).

Point 4: 254: the majority of the students remained their use stable low (in ..?)

Response 4: The sentence has been restructured (lines 254-256) to make it more clear that the majority of the students were non-users both before and during the first COVID-19 lockdown, referred to as “stable low” (which has been explained in method section 2.2.1.2 Model B – Non-users before COVID-19). “For all three substances” indicates that this applies to all three investigated substances (line 254).